# Enhancing blood availability in Latin America: A study on public perceptions and barriers to blood donation in Guatemala during the COVID-19 pandemic

**Carolina Torres Perez-Iglesias**[1]*, **Jose C. Monzon**[2], **Isabella Faria**[1], **Shreenik Kundu**[1], **Ahsan Zil-E-Ali**[3], **Nakul Raykar**[1], **Sabrina Asturias**[4]

**1** Program in Global Surgery and Social Change, Harvard Medical School, Boston, Massachusetts, United States of America, **2** Universidad Rafael Landivar, Facultad de Ciencias de la Salud, Guatemala City, Guatemala, **3** Division of Vascular Surgery, Penn State University, Hershey, Pennsylvania, United States of America, **4** Universidad Francisco Marroquin, Facultad de Medicina, Guatemala City, Guatemala

* ctorresp@bidmc.harvard.edu

## Abstract

This study explores public perceptions and the barriers to voluntary blood donation during the first year of the COVID-19 pandemic in Guatemala, a country with one of the lowest voluntary donation rates in Latin America. We additionally aimed to identify the population factors influencing blood donation behavior and to inform strategies for enhancing blood availability in the region. Between August and September 2020, an anonymous cross-sectional survey was conducted using purposive sampling. Respondents were asked about their donation history, awareness of donation processes, preferences, and barriers and motivators for blood donation. Data collection included quantitative and qualitative data. Comparative analyses by gender, age, and education level were performed and regression models were used to identify predictors of blood donation behavior. Thematic content analysis was applied to open-ended responses. Among 1141 respondents, 53.5% reported a history of previous blood donation, with the majority occurring via referred donations to family or friends (78.5%). Key factors such as male gender, older age, and higher education correlated with a higher likelihood of previous donation. Familiarity with donation centers and willingness to donate strongly influenced donation behavior. Despite 89% of never donors expressing willingness to donate, barriers like limited access to donation centers, insufficient information, and concerns over hygiene and safety were prevalent. This study highlights the significant public willingness to donate blood in Guatemala, but identified key barriers that must be addressed. Understanding these factors is essential for developing targeted initiatives to improve blood availability in Guatemala and across Latin America, particularly during health crises, such as the COVID-19 pandemic,.

provided the original author and source are credited.

**Data availability statement:** The dataset generated and/or analyzed during the current study has been uploaded as supplementary information in this submission

**Funding:** No funding was received for this manuscript.

**Competing interests:** The authors have no competing interests.

## Introduction

Ensuring a sufficient and safe supply of blood products is essential for public health, particularly in low- and middle-income countries (LMICs), where blood transfusions are critical for addressing conditions such as trauma, maternal-neonatal health, and nutritional deficiencies [1]. Many Latin American countries face a chronic shortage of voluntary blood donations, which limits their ability to meet healthcare demands. Over the last decade, voluntary blood donation rates in Latin America and the Caribbean have continued to decline, with minimal research conducted to explore the underlying causes of this trend [2]. The COVID-19 pandemic has further strained these blood systems, exacerbating already low donation rates [3].

Historically, countries in Latin America have depended on replacement-based donation systems, wherein patients must find donors through their social networks or paid donations [4,5]. This system, compounded by cultural and logistical barriers faced by the population to access donation centers, perpetuates misconceptions and limits the development of a more altruistic, voluntary blood donation culture. The COVID-19 pandemic has heightened concerns over the safety of blood donation, complicating efforts to ensure an adequate supply of blood products and further diminishing donation rates. Given the unmet demand for blood products in the region, there is an urgent need to identify the specific challenges that serve as barriers to voluntary blood donation and to implement targeted strategies to address them.

This study aimed to explore the factors that contributed to the limited availability of blood products in Guatemala, a country with the second-lowest voluntary blood donation rate in Latin America [6]. By focusing on public perceptions and barriers to donation during the first year of the COVID-19 pandemic, we also aimed to identify key demographic and behavioral factors that can inform targeted strategies to enhance blood availability in the region. Addressing these barriers is crucial for strengthening the region's healthcare systems and ensuring equitable access to life-saving blood products.

## Methods

### Study design

By 2020, the population of Guatemala exceeded 16 million people. However, voluntary blood donation rates have remained below 10% over the last decade [6]. To investigate this problem, a cross-sectional survey of the population in Guatemala was conducted between August 20th and September 9th, 2020, by DonaGuate, a non-profit organization committed to promoting a positive culture of blood donation in Guatemala [7]. This organization connects those needing blood with willing donors, creating a structured network to facilitate quick responses during emergencies. Through their platform, individuals can register to become part of the Donor network or participate in organized blood donation campaigns. DonaGuate also works closely with hospitals and other local organizations to run campaigns and educate the public about the importance of blood donation. As part of their efforts, they seek to identify and address the barriers to donation by offering resources and support to make blood donation more accessible and efficient in Guatemala. With this in mind, the survey was designed to identify public perceptions, motivators, and barriers to voluntary blood donation during the first year of the COVID-19 pandemic. Previous donor status referred to participants who reported donating blood at least once prior to the survey. The survey was anonymous and administered electronically. It comprised an 11-question instrument in Spanish that featured a combination of multiple-choice, drop-down, and open-ended questions. The translated version of the survey in English can be found in Appendix 1.

## Survey distribution

The survey was disseminated through a snowball sampling method through DonaGuate's social media platform and mailing lists. Those who received the survey via this methodology were asked to disseminate the study with their contacts. Due to this methodology, a response rate could not be estimated. Participation was entirely voluntary, and no form of compensation was provided. The survey was aimed at all Guatemalan residents over the age of 18. Incomplete responses, individuals under 18, and those residing outside of Guatemala at the time of the survey were excluded from the study. Despite the study being written in Spanish, the official language of Guatemala, we acknowledge that responses from a significant proportion of the population who does not speak Spanish could not be accessed via this survey.

## Statistical analysis

Statistical analysis was conducted using Stata 16.1 statistical software [8]. The association between study groups for the categorical variables was assessed using Pearson's Chi-Square test, which compared the observed and expected responses. The results, absolute percentages, and the count of participants in each category are presented. A multivariate logistic regression model was constructed to identify factors linked with blood donation, including only variables with a p-value <0.05 from the univariate analysis. All the variables analyzed in both the univariate and multivariate regression models were reported with unadjusted and adjusted odd ratios, respectively, along with corresponding 95% confidence intervals. Statistical significance was set at $p < 0.05$. The qualitative data from the open-ended question underwent thematic content analysis and was reviewed by two independent coders (CT, IF). Frequencies of these identified themes are also presented.

## Ethics statement

This study involved the collection of data through an anonymous survey distributed to members of the population. The survey was carefully designed and approved by DonaGuate, as part of their mission to promote blood donation in Guatemala. It ensured that no personally identifiable information was collected from participants. As such, the responses could not be traced back to any individual, thereby maintaining complete anonymity.

Given the nature of the data collection and the measures taken to ensure anonymity, the study was deemed to pose minimal risk to participants. The survey focused on gathering non-sensitive information, and there was no potential harm or distress to participants. Following these considerations, it was determined that the study did not meet the criteria for requiring Institutional Review Board approval. This determination was conducted by the Harvard Longwood Campus Institutional Review Board under protocol IRB 23-0135.

## Inclusivity in global research

Additional information regarding the ethical, cultural, and scientific considerations specific to inclusivity in global research is included in the Supporting Information (See S1 Checklist).

## Results

### Characteristics of respondents

During the study period, 1141 responses were gathered, primarily from an urban sample of 62.5% (n = 713) of females and 37.5% (n = 428) of males. From these, 94% (n = 1073) resided in Guatemala City at the time of the survey. Two-thirds of the sample were under 40, and over 80% (n = 955) reported a college education or higher. Further demographic details of the respondents are provided in Table 1.

**Table 1. Comparison of baseline characteristics and responses based on previous donation status.**

| Variables | N=1,141 | Has previously donated  N = 610 | Has never donated N = 531 | p-value |
|---|---|---|---|---|
| **Gender** | | | | |
| Male | 428 (37.5) | 271 (44.4) | 157 (29.6) | **<0.001** |
| Female | 713 (62.5) | 339 (55.6) | 374 (70.4) | |
| **Age** | | | | |
| <=29 | 438 (38.3) | 214 (35.1) | 224 (42.2) | **0.001** |
| 30-39 | 318 (27.9) | 159 (26.1) | 159 (29.9) | 0.172 |
| 40-49 | 151 (13.2) | 86 (14.1) | 65 (12.2) | 0.394 |
| 50-59 | 133 (11.7) | 79 (12.9) | 54 (10.2) | 0.163 |
| 60+ | 101 (8.9) | 72 (11.8) | 29 (5.5) | **<0.001** |
| **Educational level** | | | | |
| Elementary School/ High School | 186 (16.3) | 91 (14.9) | 95 (17.9) | **0.001** |
| Bachelor | 550 (48.2) | 273 (44.8) | 277 (52.1) | |
| Master/ Doctorate | 405 (35.5) | 246 (40.3) | 159 (30) | |
| **Place of residence** | | | | |
| Guatemala City | 1,073 (94) | 573 (93.9) | 500 (94.2) | 0.907 |
| Other | 68 (6) | 37 (6.1) | 31 (5.8) | |
| **Do you know where you can donate blood in Guatemala?** | | | | |
| No | 249 (21.8) | 87 (14.2) | 162 (30.5) | **<0.001** |
| Yes | 892 (78.2) | 523 (85.8) | 369 (69.5) | |
| **If you were to donate blood, do you care who receives your blood?** | | | | 0.032 |
| No | 1,003 (87.9) | 548 (89.8) | 455 (85.7) | |
| Yes | 138 (12.1) | 62 (10.2) | 76 (14.3) | |
| **Are you willing to donate blood?** | | | | 0.001 |
| I am not interested in donating blood | 93 (8.2) | 35 (5.7) | 58 (10.9) | |
| Yes, but only for emergencies – it does not matter who receives the blood | 256 (22.4) | 135 (22.1) | 121 (22.8) | |
| Yes, but only for emergencies of family, friends, and people I know | 281 (24.6) | 139 (22.8) | 142 (26.7) | |
| Yes, I would like to be a frequent donor (1 time per year) | 511 (44.8) | 301 (49.4) | 210 (39.6) | |

Among the respondents, 53.5% (n = 610) reported a history of blood donation. Additionally, 78.2% (n= 892) reported knowing where to donate blood, and 87.9% (n = 1,003) indicated no preference for their blood to be received by a specific individual, such as family or friends, and were willing for it to be used for anyone in need. Only 8.2% (n = 93) were not inclined to donate, and 44.8% (n = 511) were interested in becoming frequent donors (at least once annually).

Comparing donor status (previous donors vs. never donors), male respondents exhibited a higher rate of blood donation compared to females (p < 0.001, see Table 1). Respondents under 29 years of age were less commonly previous donors compared to other age groups (p = 0.001), whereas those over 60 were more likely to be previous donors (p < 0.001). Furthermore, a higher education level correlated with a greater likelihood of being a prior blood donor (p = 0.001). Never donors were less likely to know where to donate (p < 0.001) and expressed more concern about who receives the donated blood (p = 0.032).

Among previous donors (n = 610), the most common avenues for donation included referrals to family or friends (77.8%, n = 475), donations via Red Cross campaigns (10%, n = 61), donations via DonaGuate campaigns (3.3% n = 20) and other (8.8%, n = 54). Nearly half of the participants reported never having donated before the survey (n = 531). Interestingly, within the group of never donors, 89.1% (n = 473) expressed a willingness to donate in the future. Among them, 39.5% (n = 210) expressed a commitment to do so regularly (at least once per year), 26.7% (n = 142) would consider it in emergencies for family and friends, and 22.8% (n = 121) would do so in emergencies regardless of the recipient.

A regression analysis was performed to identify the factors associated with blood donation in Guatemala (Table 2). The findings revealed that being female was associated with a 58% lower likelihood of blood donation (aOR:0.42 [0.32,0.55]; p < 0.001). Older age also emerged as having a significant association with donation, as the highest odds of previous donation were observed in those aged 60 or older (aOR: 3.59 [2.14, 6.04]; p < 0.001)]. Additionally,

**Table 2. Univariate and Multivariate Regression Analyses to identify the factors associated with Blood Donation.**

|  | Unadjusted OR (95% CI) | p-value | Adjusted OR (95% CI) | p-value |
|---|---|---|---|---|
| **Gender** | | | | |
| Male | Reference | | | <0.001 |
| Female | 0.53 (0.41 - 0.67) | <0.001 | 0.42 (0.32 - 0.55) | |
| **Age** | | | | |
| 19-29 | Reference | | | |
| 30-39 | 1.05 (0.78-1.40) | 0.757 | 1.07 (0.77-1.49) | 0.668 |
| 40-49 | 1.38 (0.95 - 2.01) | 0.087 | 1.57 (1.05-2.35) | 0.029 |
| 50-59 | 1.53 (1.03 - 2.27) | 0.034 | 1.76 (1.14-2.71) | 0.01 |
| 60+ | 2.59 (1.62 - 4.15) | <0.001 | 3.59 (2.14-6.04) | <0.001 |
| **Educational level** | | | | |
| Elementary School/High School | Reference | | | |
| Bachelor | 1.03 (0.74 - 1.43) | 0.867 | 0.98 (0.69-1.41) | 0.934 |
| Master/ Doctorate | 1.62 (1.14 - 2.29) | 0.007 | 1.52 (1.03-2.24) | 0.035 |
| **Place of residence** | | | | |
| Guatemala City | Reference | | | |
| Other | 1.04 (0.64 - 1.70) | 0.871 | | |
| **Do you know where you can donate blood in Guatemala?** | | | | |
| No | Reference | | | <0.001 |
| Yes | 2.64 (1.97 - 3.54) | <0.001 | 2.67 (1.96 - 3.67) | |
| **If you were to donate blood, do you care who receives your blood?** | | | | |
| No | Reference | | | 0.756 |
| Yes | 0.67 (0.47 - 0.97) | 0.033 | 0.94 (0.62 - 1.42) | |
| **Are you willing to donate blood?** | | | | |
| I am not interested in donating blood | Reference | | | |
| Yes, but only for emergencies – it does not matter who receives the blood | 1.84 (1.13-3.01) | <0.001 | 2.31 (1.37-3.91) | 0.002 |
| Yes, but only for emergencies of family, friends, and people I know | 1.62 (1.00-2.62) | 0.048 | 2.13 (1.27-3.61) | 0.004 |
| Yes, I would like to be a frequent donor (1 time per year) | 2.37 (1.50-3.74) | <0.001 | 3.74 (2.25-6.21) | <0.001 |

individuals with higher education levels (Master's or Doctorate) demonstrated a higher rate of previous blood donation (aOR: 1.52 [1.03, 2.24]; p = 0.035). Respondents who knew where to donate and those expressing willingness to donate were more likely to have donated blood (aOR: 2.67 (1.96 - 3.67); p < 0.001 and aOR: 3.74 (2.25-6.21); p < 0.001, respectively).

## Barriers and facilitators to blood donation

The survey asked the participants what would facilitate or motivate them to donate blood. A total of 1120 open-text responses were received and analyzed using thematic analysis to identify the most common topics reported by participants. The most common themes are represented in Fig 1 (never donors) and Fig 2 (previous donors). Among both, never and previous donors, improved access to donation centers, specifically an increased number of facilities, which are easily accessible and have extended hours of operation, emerged as the most cited facilitator/motivator. The second most frequently cited motivator/facilitator by participants was the desire to help others in need, followed by the need for improved access to information about the blood donation process, including details about the procedure, duration, pre- and post-procedure requirements, recovery time, and side effects. Despite these challenges, respondents indicated an altruistic desire to help others as the strongest motivator for donation. Additional challenges identified from survey responses included the need for a more efficient and expedited donation process, increased information on how donated blood is utilized, a lack of incentives for donation, the need for attentive and respectful staff at donation centers, as well as logistical assistance coordinating appointments and reminders for future donations.

Among those who had never donated, the most cited barriers to blood donation included medical contraindications (especially age restrictions and history of transfusion-transmissible infections), unwillingness to donate blood to strangers, and limited operating hours of donation centers. Additional reported barriers can be found in Fig 3. A smaller sample of respondents also expressed hesitancy regarding the proper utilization of donated blood, expressing apprehension about potential illegal sales or unequal distribution among the population.

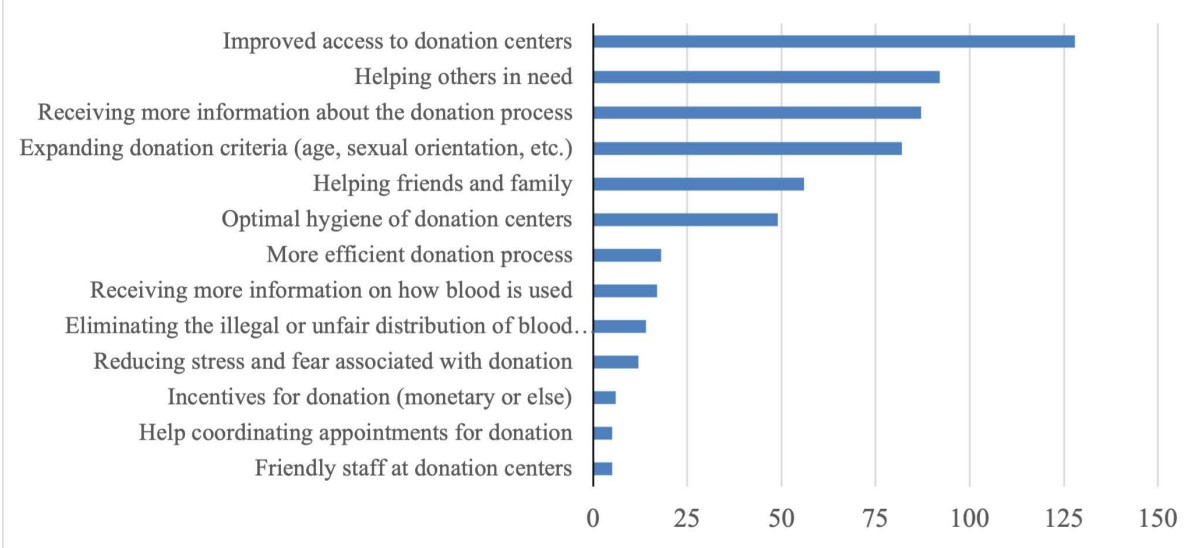

**Fig 1. Most commonly reported facilitators and motivators for blood donation among never donors.**

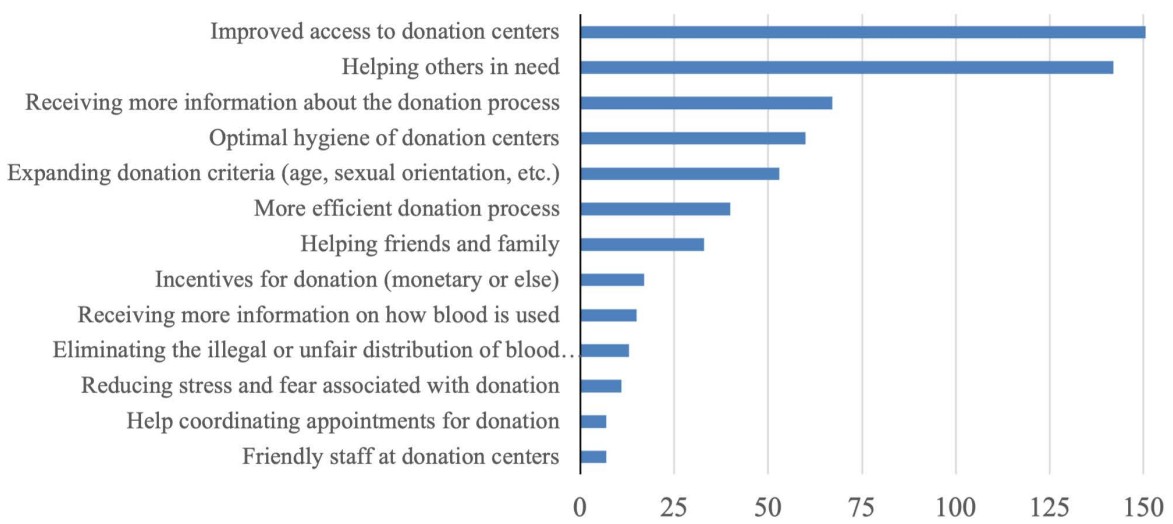

**Fig 2. Most commonly reported facilitators and motivators for blood donation among previous.**

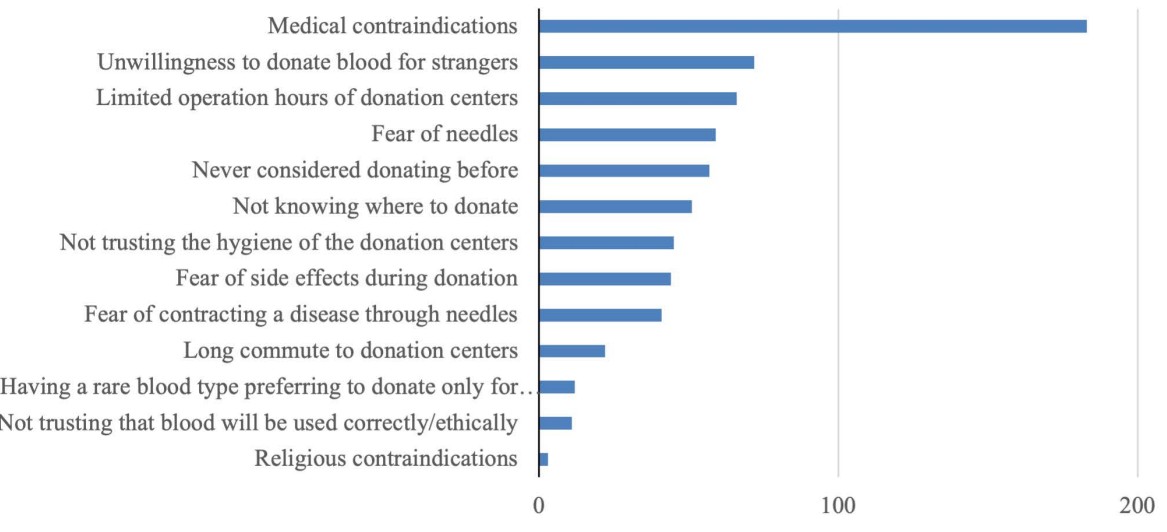

**Fig 3. Most commonly reported barriers to donation among never donors.**

## Discussion

This study provides crucial insights into the factors influencing voluntary blood donation in Guatemala, a country facing one of the lowest blood donation rates in Latin America. Among the respondents, men, older individuals, and those with higher education were significantly more likely to have previously donated blood. In contrast, younger individuals and women were less likely to have done so, reflecting a demographic gap that may help explain Guatemala's overall low donation rate, where almost half of the country's population is female. The survey also revealed that 89.1% of respondents who had never donated blood

expressed a willingness to do so in the future, highlighting the potential for increasing voluntary donations. To achieve this goal, key barriers identified from this population group, such as limited access to centers, campaigns, and information regarding blood donation, need to be addressed in the region.

One such barrier is the accessibility of donation centers to the general population. With only 37 transfusional centers serving a population of over 17 million, most of which are located in central regions of the country, logistical difficulties, such as long travel distances and restricted operating hours, prevent many potential donors from participating [9]. Expanding the network and accessibility of donation centers is particularly critical given that those who knew where to donate were significantly more likely to have donated in the past and can potentially educate others on how to do so.

Another essential barrier found in our study was the need for more information about the donation process and its safety, particularly among never donors. Notably, declining voluntary blood donation rates have been described in Latin America and the Caribbean during the COVID-19 pandemic [10]. In Guatemala, this rate decreased from 4.1% to 3.7% in 2020 [10]. An important percentage of respondents cited hygienic concerns at donation centers as their primary barrier to voluntary donation. During the time the survey was conducted, the country was in a critical phase of the COVID-19 pandemic, experiencing significant restrictions, such as curfews, mandatory mask mandates, limited public transportation, and restrictions on public gatherings. Many healthcare services were also focused on pandemic-related care, which may have influenced general access to and attitudes towards blood donation. While some restrictions have since been lifted, the long-term consequences of the pandemic remain relevant to understanding the local healthcare context and perceptions of the general population. Moving forward, we believe there is a need for better public education and outreach campaigns to provide clear, accurate information about the donation process and its safety, in addition to fortifying protective measures for the population. Tapping into the motivation of the population to help others in need through targeted messaging and education on the effects of blood donation could be highly effective. Targeted educational campaigns focused on dispelling common misconceptions about blood donation and providing accurate information about the safety, risks, and life-saving potential are critical to promoting a positive culture of voluntary blood donation in the country. This will be critical not only in response to the current low donations rates and the COVID-19 pandemic, but also an important strategy in preparation for potential future health emergencies [10,11].

Interestingly, a notable proportion of respondents, particularly those who had never donated blood, expressed concerns about the ethical use of donated blood. Fears about the potential misuse or illegal sale of blood products reflect broader societal mistrust in public institutions, a known issue in regions with high levels of corruption and weak governance. Building trust in the blood donation system will be essential to overcoming these barriers. Public health authorities and non-profit organizations, such as DonaGuate, have a critical role in ensuring transparency in the management and use of blood products. Efforts to improve transparency and accountability, coupled with campaigns that build trust and confidence in the donation system, could help alleviate these concerns and encourage more people to donate. Lastly, improving the efficiency of the blood donation process itself, such as reducing wait times, facilitating appointments, and appointing attentive, empathetic staff could enhance the donor experience and increase the likelihood of repeat donations. Moreover, providing logistical support, such as appointment reminders and transportation assistance, could further reduce barriers to participation.

The challenges identified in Guatemala align with those encountered in other low- and middle-income countries [12–16]. Particularly in Latin America, countries like Argentina,

Costa Rica, Chile, and Nicaragua have implemented successful strategies to improve the availability of blood products, shifting from a replacement-based model to an altruistic donation model, primarily through the reorganization of the structure of the blood collection system. The implementation of the Regional Plan of Action for Transfusion Safety developed by the Pan American Health Organization (PAHO), which strategies include the planning and management of the national blood network system, the promotion of voluntary blood donation, the quality assurance, and the appropriate use of blood and blood components, helped increase the voluntary blood donation rate from 38% to 100% in Nicaragua [17]. Chile increased altruistic donation from 8% to 22% by reorganizing autonomously functioning blood collection centers into transfusion centers, modernizing the data collection systems, and implementing mobile blood collection campaigns [18,19]. Mobile blood collection units have proven successful in increasing voluntary donation rates in other LMICs and could be particularly effective in Guatemala, where geographical barriers significantly limit access to donation centers [20]. These initiatives will require the commitment and leadership of health authorities to integrate these improvements into the national health system [21]. The government's commitment to recognizing the importance of blood transfusion and supporting the health care system with the appropriate infrastructural, human, and financial resources is indispensable for achieving these goals [22]. The design of culturally sensitive and demographically tailored approaches can positively impact the population [23,24]. Furthermore, creating a safe and welcoming atmosphere where donors feel at ease and confident that their requirements during the donation process are being addressed can encourage a willingness to return for future donations [25].

Lastly, community engagement and partnerships with governmental and civil organizations, such as Non-Governmental Organizations, religious institutions, and societies, are necessary to implement tailored policies that support voluntary donation. Reviewing and updating the national guidelines for donor eligibility and blood safety standards to reflect evidenced-based strategies that support voluntary blood donation is critical to improving the voluntary blood donation rate [26]. The PAHO currently recommends an upper limit age for blood donation that varies between 65 and 81 years based on the health conditions of the local donor population [27]. Currently in Guatemala, only those below 55 years of age are allowed to donate blood not more often than every three months for males and four months for females. The reduction in the interval of time allowed between donations as far as every 8 weeks in males and every 12 weeks in females has also proven effective in increasing the blood product supply without significantly affecting quality of life or physical activity [28]. In the post-pandemic era, modifying eligibility criteria for donation emerges as an essential strategy to broaden inclusivity for blood donors. Other nations have implemented this measure through partnerships with civil societies by removing questions related to potential high-risk behaviors during donor screening to remove the stigma associated with these and potentially tap into a broader pool of willing donors [29].

There are important limitations to our study. This electronic survey was distributed through purposive sampling by a non-profit organization in Guatemala, which may have attracted individuals with a higher intention or willingness to donate blood. Due to this methodology, we are also unable to calculate a response. In addition, despite the broad distribution of the study instrument, most respondents were located in Guatemala City, which impedes the generalizability of the results. Nonetheless, our results are still relevant to the region and can serve as a foundational framework for future research to comprehensively increase voluntary blood donation rates.

In conclusion, this study sheds light on the complex interplay of demographic, logistical, and cultural factors that contribute to the low blood donation rates in Guatemala. Addressing these challenges will require a multi-faceted approach that includes expanding access to donation centers, improving public education, building trust in the donation system, and enhancing the efficiency of the donation process. The findings of this study offer a roadmap for policymakers and public health officials to develop targeted interventions that address the specific barriers to blood donation in Guatemala, helping to ensure a more robust and reliable blood supply for those in need, particularly during crises like the COVID-19 pandemic.

## Supporting information

**S1 Checklist. Inclusivity in global research.**
(DOCX)

**S1 Data. Dataset.**
(XLSX)

**S1 Appendix. Translated survey instrument in English.**
(DOCX)

## Acknowledgments

The authors would like to acknowledge DonaGuate (https://www.donaguate.com) and their commitment to promoting a positive culture of blood donation in Guatemala.

## Author contributions

**Conceptualization:** Carolina Torres Perez-Iglesias, Jose Monzon, Nakul Raykar, Sabrina Asturias.

**Data curation:** Carolina Torres Perez-Iglesias, Jose Monzon, Isabella Faria, Shreenik Kundu, Ahsan Zil-E-Ali, Nakul Raykar, Sabrina Asturias.

**Formal analysis:** Carolina Torres Perez-Iglesias, Isabella Faria, Shreenik Kundu, Ahsan Zil-E-Ali.

**Investigation:** Carolina Torres Perez-Iglesias, Jose Monzon, Isabella Faria, Shreenik Kundu, Ahsan Zil-E-Ali, Nakul Raykar, Sabrina Asturias.

**Methodology:** Carolina Torres Perez-Iglesias, Isabella Faria, Shreenik Kundu, Ahsan Zil-E-Ali, Nakul Raykar, Sabrina Asturias.

**Project administration:** Carolina Torres Perez-Iglesias, Jose Monzon, Shreenik Kundu, Nakul Raykar, Sabrina Asturias.

**Resources:** Carolina Torres Perez-Iglesias, Jose Monzon, Sabrina Asturias.

**Software:** Ahsan Zil-E-Ali.

**Supervision:** Carolina Torres Perez-Iglesias, Nakul Raykar, Sabrina Asturias.

**Validation:** Carolina Torres Perez-Iglesias, Jose Monzon, Nakul Raykar, Sabrina Asturias.

**Visualization:** Carolina Torres Perez-Iglesias, Nakul Raykar.

**Writing – original draft:** Carolina Torres Perez-Iglesias, Jose Monzon, Isabella Faria, Ahsan Zil-E-Ali.

**Writing – review & editing:** Carolina Torres Perez-Iglesias, Jose Monzon, Isabella Faria, Shreenik Kundu, Nakul Raykar, Sabrina Asturias.

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
