## [Decision Letter · Decision Letter 0]

13 Aug 2024

PGPH-D-24-01168

Enhancing Blood Availability in Latin America: A Study on Public Perceptions and Barriers to Blood Donation in Guatemala during the COVID-19 Pandemic

Dear Dr. Torres Perez-Iglesias,

Thank you for submitting your manuscript to PLOS Global Public Health. After careful consideration, we feel that it has merit but does not fully meet PLOS Global Public Health’s publication criteria as it currently stands. Therefore, we invite you to submit a revised version of the manuscript that addresses the points raised during the review process.

Please note that we have only been able to secure a single reviewer to assess your manuscript. We are issuing a decision on your manuscript at this point to prevent further delays in the evaluation of your manuscript. Please be aware that the editor who handles your revised manuscript might find it necessary to invite additional reviewers to assess this work once the revised manuscript is submitted. However, we will aim to proceed on the basis of this single review if possible. 

We look forward to receiving your revised manuscript.

Kind regards,

Joanna Tindall

Staff Editor

Journal Requirements:

2. In the online submission form, you indicated that The datasets generated and/or analyzed during the current study are available from the corresponding author on reasonable request. 

a. In a public repository, 

b. Within the manuscript itself, or 

c. Uploaded as supplementary information.

Additional Editor Comments (if provided):

Reviewers' comments:

Reviewer's Responses to Questions

**Comments to the Author**

1. Does this manuscript meet PLOS Global Public Health’s publication criteria ? Is the manuscript technically sound, and do the data support the conclusions? The manuscript must describe methodologically and ethically rigorous research with conclusions that are appropriately drawn based on the data presented.

Reviewer #1: No

2. Has the statistical analysis been performed appropriately and rigorously?

Reviewer #1: Yes

3. Have the authors made all data underlying the findings in their manuscript fully available (please refer to the Data Availability Statement at the start of the manuscript PDF file)?

Reviewer #1: Yes

4. Is the manuscript presented in an intelligible fashion and written in standard English?

Reviewer #1: Yes

5. Review Comments to the Author

Reviewer #1: Thank you for your interesting work on blood donation in Guatemala and barriers to donate. Attached you will find some suggestions that might help you with your revision.

- Introduction: The title of the essay, the abstract, and the introduction do not seem to be well-coordinated. What exactly is the goal of the analysis? Please clarify this point further in the abstract and introduction

- Introduction: I miss seeing more literature on the motivations and barriers of donors from other Latin American countries in the introduction. Isn't there a larger body of research on this topic?

- Introduction: "Moreover, these nations commonly grapple with a range of challenges that contribute to the limited availability of blood products…” The list in this sentence could be clearer with semicolons or separate points.

- Introduction: “urgent urgency” may be simplified to “urgent need”

- Methods: It would greatly help the readers to know more about the survey. How many people were contacted? What were the age limits for participation? Did all population groups have access to the survey? Were there any population groups that could not participate due to a language barrier? What was the response rate of the survey? To what extent does the sample accurately reflect the population? If no information is available on this, it would still be helpful to know how the participants were selected.

- Methods: What does it mean if someone has already donated? When was this donation? Were these whole blood donations? How exactly was the question phrased?

- Methods: I am a bit confused. Under 'statistical analysis,' it states that a review board was involved. Under 'ethics statements,' it says that no board was necessary. Please clarify this discrepancy.

- Methods: Could you please give more information on Dona Guate?

- Results: Could you explain in more detail what you mean by ‘preference regarding the recipient of their donated blood’

- Results: “Comparing donor status (previous donors vs. never donors), male respondents exhibited a higher rate of blood donation compared to females (p < 0.001). Respondents under 29 years of age were less commonly previous donors compared to other age groups (p = 0.001), whereas those over 60 were more likely to be previous donors (p < 0.001)“ Could you please explain on which table you refer here?

- Results: “Among previous donors (n = 610), the most common avenues for donation included referrals to family or friends (77.8%, n = 475), donations via Red Cross campaigns (10%, n = 61),…” Could you please explain on which table you refer here?

- Results: could you please explain how ‘improved access to donation centers’ could be a motivation to donate? How did you ask about motivation?

- Results: access to information about blood donation is the third and not the second most frequently mentioned facilitator.

- Discussion: Please include a section at the beginning of the discussion summarizing the key findings of the analysis. What are the most important results?

- Discussion: Please give information on source 29

- In the discussion, there is little reference to the results of the survey. What are the conclusions drawn from the presented analysis? Why are blood donation rates particularly low in Guatemala?

- Discussion: In the limitations section, it should be noted that most of the respondents were from Guatemala city.

- Figure 1 and 2: It would ne helpfull to see the values in percent.

6. PLOS authors have the option to publish the peer review history of their article (what does this mean? ). If published, this will include your full peer review and any attached files.

**Do you want your identity to be public for this peer review?** For information about this choice, including consent withdrawal, please see our Privacy Policy .

Reviewer #1: **Yes: ** Christian Weidmann

---

## [Decision Letter · Decision Letter 1]

14 Nov 2024

PGPH-D-24-01168R1

Enhancing Blood Availability in Latin America: A Study on Public Perceptions and Barriers to Blood Donation in Guatemala during the COVID-19 Pandemic

Dear Dr. Torres Perez-Iglesias,

Thank you for submitting your manuscript to PLOS Global Public Health. After careful consideration, we feel that it has merit but does not fully meet PLOS Global Public Health’s publication criteria as it currently stands. Therefore, we invite you to submit a revised version of the manuscript that addresses the points raised during the review process.

The manuscript has been evaluated by two reviewers, and their comments are available below.

One reviewer still has major concerns. They feel the manuscript would benefit from improvements to the reporting of methodological aspects of the study, and has raised several questions regarding the survey. 

Could you please carefully revise the manuscript to address all comments raised?

We look forward to receiving your revised manuscript.

Kind regards,

Johanna Pruller, Ph.D.

PLOS Staff Editor

Journal Requirements:

Additional Editor Comments (if provided):

Reviewers' comments:

Reviewer's Responses to Questions

**Comments to the Author**

1. If the authors have adequately addressed your comments raised in a previous round of review and you feel that this manuscript is now acceptable for publication, you may indicate that here to bypass the “Comments to the Author” section, enter your conflict of interest statement in the “Confidential to Editor” section, and submit your "Accept" recommendation.

Reviewer #1: All comments have been addressed

Reviewer #2: (No Response)

2. Does this manuscript meet PLOS Global Public Health’s publication criteria ? Is the manuscript technically sound, and do the data support the conclusions? The manuscript must describe methodologically and ethically rigorous research with conclusions that are appropriately drawn based on the data presented.

Reviewer #1: Yes

Reviewer #2: Yes

3. Has the statistical analysis been performed appropriately and rigorously?

Reviewer #1: Yes

Reviewer #2: Yes

4. Have the authors made all data underlying the findings in their manuscript fully available (please refer to the Data Availability Statement at the start of the manuscript PDF file)?

Reviewer #1: Yes

Reviewer #2: Yes

5. Is the manuscript presented in an intelligible fashion and written in standard English?

Reviewer #1: Yes

Reviewer #2: Yes

6. Review Comments to the Author

Reviewer #1: (No Response)

Reviewer #2: (No Response)

7. PLOS authors have the option to publish the peer review history of their article (what does this mean? ). If published, this will include your full peer review and any attached files.

**Do you want your identity to be public for this peer review?** For information about this choice, including consent withdrawal, please see our Privacy Policy .

Reviewer #1: **Yes: ** Christian Weidmann

Reviewer #2: No

---

## [Decision Letter · Decision Letter 2]

16 Jan 2025

Enhancing Blood Availability in Latin America: A Study on Public Perceptions and Barriers to Blood Donation in Guatemala during the COVID-19 Pandemic

PGPH-D-24-01168R2

Dear Dr Torres Perez-Iglesias,

We are pleased to inform you that your manuscript 'Enhancing Blood Availability in Latin America: A Study on Public Perceptions and Barriers to Blood Donation in Guatemala during the COVID-19 Pandemic' has been provisionally accepted for publication in PLOS Global Public Health.

Best regards,

Julia Robinson

Executive Editor

Reviewer Comments (if any, and for reference):

Reviewer's Responses to Questions

**Comments to the Author**

1. If the authors have adequately addressed your comments raised in a previous round of review and you feel that this manuscript is now acceptable for publication, you may indicate that here to bypass the “Comments to the Author” section, enter your conflict of interest statement in the “Confidential to Editor” section, and submit your "Accept" recommendation.

Reviewer #2: All comments have been addressed

2. Does this manuscript meet PLOS Global Public Health’s publication criteria ? Is the manuscript technically sound, and do the data support the conclusions? The manuscript must describe methodologically and ethically rigorous research with conclusions that are appropriately drawn based on the data presented.

Reviewer #2: Yes

3. Has the statistical analysis been performed appropriately and rigorously?

Reviewer #2: N/A

4. Have the authors made all data underlying the findings in their manuscript fully available (please refer to the Data Availability Statement at the start of the manuscript PDF file)?

Reviewer #2: Yes

5. Is the manuscript presented in an intelligible fashion and written in standard English?

Reviewer #2: (No Response)

6. Review Comments to the Author

Reviewer #2: (No Response)

7. PLOS authors have the option to publish the peer review history of their article (what does this mean? ). If published, this will include your full peer review and any attached files.

**Do you want your identity to be public for this peer review?** For information about this choice, including consent withdrawal, please see our Privacy Policy .

Reviewer #2: No
